# Identification of Adolescents with Adiposities and Elevated Blood Pressure and Implementation of Preventive Measures Warrants the Use of Multiple Clinical Assessment Tools

**DOI:** 10.3390/jpm11090873

**Published:** 2021-08-31

**Authors:** Hiba Bawadi, Manal Kassab, Abdel Hadi Zanabili, Reema Tayyem

**Affiliations:** 1Department of Health Sciences, Qatar University, Doha P.O. Box 2713, Qatar; hbawadi@qu.edu.qa; 2Faculty of Nursing, Jordan University of Science & Technology (JUST), P.O. Box 3030, Irbid 22110, Jordan; manal_kassab@yahoo.com; 3Department of Nutrition and Food Technology, Jordan University of Science and Technology, P.O. Box 3030, Irbid 22110, Jordan; a.hady.zanabili@hotmail.com

**Keywords:** abdominal adiposity, adolescents, blood pressure, obesity, Jordan, boys

## Abstract

The burden of abdominal adiposity has increased globally, which is recognized as a key condition for the development of obesity-related disorders among youth, including type 2 diabetes, cardiovascular disease, and hypertension. High blood pressure (BP) and cardiovascular diseases increase the rates of premature mortality and morbidity substantially. **Aims**: to investigate the relation between abdominal adiposity and elevated BP among adolescent males in Jordan. **Methods**: Nationally representative sample of male adolescents was selected using multi-cluster sampling technique. Study sample included 1035 adolescent males aged 12 to 17 years. Multiple indicators were used to assess adiposity including waist circumference (WC) and total body fat (TF), truncal fat (TrF), and visceral fat (VF). Systolic blood pressure was measured to assess hypertension. **Results**: After adjusting for age, smoking status, and physical activity, the odds of having stage two hypertension increased 6, 7, and 8 times for adolescents who were on 90th percentile or above for Trf, VF, and WC, respectively. **Conclusion**: Elevated BP was significantly associated with total and abdominal adiposity among adolescent males in Jordan. Use of multiple clinical assessment tools is essential to assess abdominal obesity among adolescents.

## 1. Introduction

Obesity is considered the most serious future-health threatening problem among children and adolescents. In Jordan, about 17.3% of children and adolescents were overweight, and 15.7% were obese, as documented by Zayed and colleagues (2016) [1]. Recent research indicates that abdominal or central obesity is a more robust predictor for several debilitating and deadly chronic diseases than overall adiposity [2,3]. Besides being considered an important predictor of adult obesity, abdominal obesity among children and adolescents is also linked to several life threatening diseases, including cardiovascular diseases, diabetes, orthopaedic problems, cancers, metabolic syndrome, and high blood pressure [4,5,6,7]. Obesity does this through a variety of pathways, some as straightforward as the mechanical stress of carrying extra pounds and some involving multipart alterations in both hormones and cellular metabolism [8].

In the last decade, elevated BP increased among children and adolescents [9]. Adolescents with elevated BP can develop several chronic diseases and body organ damage [10,11]. Children with high BMI are more prone to have elevated BP [12]. Premature mortality is attributed to elevated BP by increasing the incidence of cardiovascular disease [13]. Prevention of childhood obesity initiates reduction in BP which leads to substantial prevention of cardiovascular disease [14]. Several studies supported the association between elevated BP and obesity among children and adolescents [15,16]. In Jordan, there are not enough data available about rates of abdominal obesity among children and adolescents, nor the association between abdominal adiposity and elevated blood pressure among this vulnerable group. The limited data available in Jordan used a single assessment tool for obesity with no consideration for identification and specification of risk and precision of preventive treatment. Therefore, this study aimed to investigate the relation between abdominal adiposity and elevated BP among adolescent males in Jordan by implementing a panel of assessment tools.

## 2. Materials and Methods

### 2.1. Study Design

A population-based, nationally representative cross-sectional study was conducted, and included 1035 male adolescents from 18 elementary and high schools in Jordan across the whole kingdom of Jordan. The study protocol was examined and approved by Jordan University of Science, Technology, and Ministry of Education-Jordan.

A list of all males’ schools (341 schools) in Jordan was obtained from Ministry of Education (MOE). Three geographical zones were identified based on MOE classification; north, mid, and south Jordan. Distribution of Schools over the geographical zones was also obtained from MOE; 130 schools in north area, 159 schools in middle area, and 52 schools in south area. Based on this distribution, a sample of 5% of schools (n = 18 schools) were selected as follows: 7 schools in northern Jordan, 8 schools in mid-Jordan, and 3 schools in southern Jordan. Classes in each school and adolescents in each class were selected using simple random selection.

### 2.2. Data Collection

All adolescents were asked to complete a self-administered questionnaire including data about socio demographic status, dietary habits, and physical activity level. The questionnaire was pilot tested to check for any vague, poorly stated items.

### 2.3. Measurements

#### 2.3.1. Total Adiposity

Total adiposity was assessed using body mass index (BMI) and total body fat (TF) percentage. Anthropometric measurements were performed by a trained dietitian. Weight was measured to the nearest 0.1 kg (Tanita BF-350, Tanita Corporation, Tokyo, Japan). Height was measured to the nearest 0.1 cm with a portable Harpenden stadiometer. BMI was then calculated as the ratio of weight (kg) to height (m) squared (kg/m^2^) [17]. BMI for age charts were used to assess overweight and obesity [18]. BMI and age were plotted to obtain adolescents’ BMI-age percentiles. Data were interpreted as follows: <5th percentile were assessed as underweight, 5th–85th percentile were assessed as normal weight, 85th–95th percentile were assessed as overweight, and ≥95th percentile were assessed as obese [19].

TF% was estimated using bioelectrical impedance technique (Tanita BF-350, Tanita Corporation, Tokyo, Japan). Age and gender sensitive cut-off points for TF% were used to assess obesity [19].

#### 2.3.2. Abdominal Adiposity

Several indicators were used to assess abdominal adiposity; waist circumference (WC), truncal fat (TrF) %, and visceral fat (VF). WC was measured at the narrowest area of the torso, as seen from the anterior aspect, to the nearest 0.1 cm with a non-elastic tape measure snugly fitted to measure WC at the level of the natural waist. The WC value corresponding to ≥90th percentile of WC for gender and age were used as cut-off values to identify adolescents with abdominal obesity [20].

Truncal fat (TrF)% and VF were estimated using bioelectrical impedance technique (Tanita AB-140 ViScan; Tanita Corporation, Tokyo, Japan). Adolescents were asked to lie back and expose their abdominal region. Electrodes were attached to the impedance meter and placed across the navel of the subject with the electrodes facing down. The positioning line was aligned on the impedance meter with the laser to adjust the position. Values associated with ≥90th percentile were considered high.

### 2.4. Blood Pressure Measurement

Blood pressure (BP) was measured by a registered nurse using validated automated oscillometric device (Omron HEM-7051, Omron Corporation, Kyoto, Japan) [21]. Blood pressure was measured according to a standardized protocol. Blood pressure was measured first on the right arm and then at left arm. Both readings were taken after the participants were allowed to sit and rest for 5 min to relieve anxiety with their back supported, feet on the floor, arm supported, and cubital fossa at heart level. The same technique was followed for right and left arm measurement. Two readings were obtained at a 1-min interval, and the average of the two readings was recorded. Systolic BP was determined by the onset of the “tapping” KSs (K1) and the fifth KS (K5), or the disappearance of KS, as the definition of Diastolic BP. After measuring the mean values of BP, the BP percentiles were determined accordingly.

The following criteria were used to categorize adolescents according to their BP: normal BP for adolescents with systolic BP <90th percentile; prehypertension for adolescents with 90th to <95th percentile; stage 1 hypertension for adolescents with 95th to <99th percentile and stage 2 hypertension for adolescents with >99th percentile [22]. Only Systolic BP was used in this study as indicator for hypertension, as the literature suggests that diastolic hypertension rarely occurs without Systolic hypertension in children and adolescents [23,24]. 

### 2.5. Statistical Analysis

Statistical analysis was performed using SPSS for Windows version 11.5 (Statistical Package for the Social Sciences). Data were expressed as frequencies and percentages. Multivariate logistic regression was performed to investigate the association between adolescent’s risk of elevated systolic blood pressure and anthropometric measurements. Odds ratios (OR) and 95% confidence intervals (95% CI) were calculated and *p*-value was set at *p* < 0.05.

## 3. Results

This study included 1035 adolescents aged between 12 and 17 years. Socio-demographic characteristics of Jordanian adolescents were presented in Table 1. Half of participants were from Mid Jordan (49%), followed by North (35%), and South (16%). Participant’s school grades ranged from 7th grade up to 11th grade. Students’ parenteral educational levels were high school diplomas and college education in most cases. Few students reported either an illiterate mother or father.

Table 2 presents data related to systolic hypertension according to their dietary habits. Students’ self-reported regular consumption of vegetables (*p* = 0.008), fruits (*p* = 0.033) and chocolate (*p* = 0.013) were related to their systolic blood pressure. On the other hand, milk and dairy products, meat, fish, legumes, canned fruit juice, soda, and nuts were not significantly associated with blood pressure. Table 3 shows adjusted odds ratios for adolescents’ SHTN based on their general and abdominal obesity. After adjusting for confounding variables—age, smoking status, and physical activity—it was found that BMI, WC, TF, TrF, and VF are positively related to increased odds of SHTN.

Odds of pre-hypertension increased in a dose–response pattern among overweight (OR, 2.3; 95% CI, 1.4–3.8) and obese (OR, 2.7; 95% CI, 1.6–4.5) adolescents as compared to adolescents with normal body weight. Similarly, odds of stage 1 hypertension increased in a dose–response pattern across overweight (OR,1.8; 95% CI, 1.01–3.1) and obesity (OR, 2.7; 95% CI, 1.6–4.6) classes. A similar trend was observed for stage 2 SHTN.

With regard to WC, and after adjusting for age, smoking status, and physical activity, adolescents with the highest 10th percentile of WC had 5 times higher odds of having stage 1 hypertension (OR, 95% CI, 2.3–10.6) and 8.6 higher odds of having stage 2 hypertension (OR, 95% CI, 4,5–16.4). A similar relation was found with total body fat. Obese adolescents had twice as high odds of having stage 1 hypertension (OR, 95% CI, 1.3–3.8) and 3.5 times higher odds of having stage 2 hypertension (OR, 95% CI, 2.3–5.5).

With regard to TrF, and after adjusting for age, smoking status, and physical activity, adolescents with the highest 10th percentile of TF had 4 times higher odds of having stage 1 hypertension (OR, 95% CI, 2.2–7.1) and 5.5 higher odds of having stage 2 hypertension (OR, 95% CI, 3.4–9.1). Similar relation was found with VF. Adolescents with the highest 10th percentile of VF had 4.2 times higher odds of having stage 1 hypertension (OR, 95% CI, 1.8–9.4) and 7.2 times higher odds of having stage 2 hypertension (OR, 95% CI, 3.7–14.1).

## 4. Discussion

Health consequences of overweight and obesity in adolescence are strongly associated with risk factors for chronic health conditions such as cardiovascular disease, diabetes, and elevated blood pressure [4,6].

Fruit and vegetable consumption was significantly related to blood pressure (*p* = 0.008 and *p* = 0.033, respectively). This finding was consistent with the findings of several studies [25,26,27,28]. Apple et al. (2006) studied the contents of fruit and vegetables for vitamins, minerals, and fibers. Fruits and vegetables also contained potassium; and this increase consumption of potassium intake was associated with reduction in blood pressure. The increase in potassium intake had same lowering effect on blood pressure as a decrease in sodium intake. Potassium plays a major role in balancing out the negative effects of sodium. Whelton et al. (1997) recommended potassium for prevention and treatment of hypertension. Increasing serum levels of vitamins A, C, E [25], and D [28] were associated with lowering blood pressure. Meta-analysis suggested that increasing the dietary fiber intake had a lowering effect on blood pressure [27]. According to this study, chocolate was significantly related to blood pressure (*p* = 0.013). Studies explained chocolate’s role to lower blood pressure [29,30,31]. Chocolate contained coca that include polyphenols, especially flavones. Strong effects of flavones on blood pressure as a vasodilator were applied by increasing the formation of endothelial nitric oxide.

Findings of the current study are in line with previously published studies suggesting that obesity is an independent risk factor for early hypertension [4,6,23,24]. Abdominal adiposity showed stronger association that between fatness and the risks of stage 1 and stage 2 systolic hypertension among study participants. Risks increased up to 8 times with abdominal adiposity indicators. Our findings were similar to those reported by Pausova et al. (2012) and Pazin et al. (2020), where an association was found between adolescents abdominal adiposity and high BP, especially among in boys [32,33].

The relationship between abdominal adiposity and elevated blood pressure is not fully explained yet. However, researchers linked abdominal adiposity to elevated blood pressure via insulin resistance and related inflammation [8,24,34,35,36,37]. Abdominal adiposity stimulates the state of insulin resistance and the inflammatory process [38]. Inflammation leads several changes vascular endothelial function, which in turn leads to the development of hypertension [39]. Visceral fat that accumulates around individual internal organs decreases body sensitivity to insulin, which contributes to oxidative stress, inflammation, vascular endothelial dysfunction, and hypertension [35,37].

Landsberg et al. (2013) explained that the pathophysiology of fat accumulation in the abdominal region led to an increase in BP through the stimulation of insulin resistance, increasing central nervous system activity, renin-angiotensin-aldosterone system activity, angiotensinogen from intra-abdominal adipocytes, aldosterone production, and the renal sodium reabsorption [40]. High adiposity can lead to adiposopathy in adolescence, with associated increases in inflammation and oxidative stress, changes in adipokines and endocrinopathy [39]. This manifests as cardio-metabolic risk factors in similar patterns to those in noted in obese adults [5]. Both obesity and cardio-metabolic risk factors have been shown to be associated with vascular changes indicative of early hypertension, atherosclerosis, as well as ventricular hypertrophy, dilation, and dysfunction [8,24].

Ke and colleagues (2018) clarified that high BP is being induced by VF due to high levels of free fatty acids excretion in liver via portal circulation after lipogenesis and lipolysis activity, gluconeogenesis, lipid synthesis, and insulin resistance [41]. This elevated amount of free fatty acids will induce hypertension and eventually atherosclerosis [42,43].

Many studies revealed that applying different anthropometric assessment methods is essential to evaluate body weight status accurately [44,45,46,47]. Body composition can be assessed at the molecular, cellular, and tissue levels using several different methods [44,46]. Specific limitations of anthropometry in the obese are obvious. These limitations include the inability to distinguish subcutaneous fat from visceral adipose tissue, which is helpful to assess disease risk [47] and accuracy may be lowered in the severely obese, due to difficulties finding the actual waistline or drooping abdominal fat can interfere with hip measurement [46]. Therefore, no single measurement is recommended to assess the body composition of obese people. Yet, each modality has benefits and drawbacks [46].

This study has several strengths, including the representation of the sample to the Jordanian adolescent males and multiple tools used to operationalize adiposity. However, the study is limited as it did not include intensive dietary data which were not to feasible in the school setting, especially with all the measurements being performed.

In conclusion, excess fat, especially abdominal fat, is associated with increased risk of systolic hypertension among adolescent males. The physiological relation between obesity and hypertension could not attribute to a single factor. It is critical to use multiple clinical assessment tools for obesity to optimizes identification of adolescents at risk; hence preventive measures can be timely considered. Findings of this study must be taken with high level of importance to prevent early hypertension and associated cardiovascular diseases risk later in adulthood.

## Figures and Tables

**Table 1 jpm-11-00873-t001:** Socio-demographic characteristic of Jordanian adolescents.

Variable	*n* (%)
**Region**	
Northern Jordan	363 (35.1)
Mid-Jordan	508 (49.1)
Southern Jordan	164 (15.8)
**School grade**	
7th	181 (17.5)
8th	230 (22.2)
9th	224 (21.6)
10th	220 (21.3)
11th	180 (17.4)
**Father educational level ^†^**	
Illiterate	23 (2.2)
≤12 years	538 (52.0)
>12 years	474 (45.8)
**Mother educational level ^†^**	
Illiterate	17 (1.6)
≤12 years	576 (55.7)
>12 years	442 (42.7)
**Family Income (JD) ***	
≤300	176 (17.0)
301–499	297 (28.7)
500–799	234 (22.6)
≥800	296 (28.6)
**Cigarette smoking**	
Yes	243 (23.5)
No	792 (76.5)

* Jordanian Dinar (JD) = USD 1.408. ^†^ education level: ≤12 years: include elementary and high schools, ≥12 years: include after high school education.

**Table 2 jpm-11-00873-t002:** Prevalence of hypertension according to adolescents food group intake.

Frequency of Eating Different Food/Week	Mean ± SD	*p*-Value	Frequency of Eating Different Food/Week	Mean ± SD	*p*-Value
**Vegetable**			**Nuts**		
Daily	121.25 a ± 13.22	**0.008**	Daily	120.63 ± 13.11	**0.162**
4–6	123.78 b ± 13.38		4–6	122.90 ± 13.93	
1–3	123.08 b ± 13.29		1–3	123.38 ± 13.29	
No	128.40 c ± 16.77		No	123.91 ± 13.65	
**Fruit**			**Chocolate**		**0.013**
Daily	121.07 a ± 13.22	**0.033**	Daily	121.81 a ± 13.23	
4–6	122.79 b ± 13.18		4–6	121.07 a ± 12.66	
1–3	124.22 b ± 13.76		1–3	124.56 b ± 13.89	
No	125.21 c ± 15.28		No	124.59 c ± 14.31	
**Milk and dairy products**			**Soda**		
Daily	121.76 ± 12.89	**0.113**	Daily	122.90 ± 12.65	**0.198**
4–6	122.27 ± 13.46		4–6	122.03 ± 13.41	
1–3	124.31 ± 13.75		1–3	123.24 ± 13.71	
No	122.82 ± 14.48		No	122.59 ± 15.42	
**Meat**			**Canned fruit juice**		
Daily	121.69 ± 12.71	**0.061**	Daily	122.28 ± 12.82	**0.187**
4–6	122.55 ± 12.81		4–6	121.22 ± 13.44	
1–3	122.59 ± 13.68		1–3	123.05 ± 13.64	
No	126.36 ± 15.70		No	125.03 ± 15.29	
**Fish**			**Legumes**		
Daily	122.04 ± 18.14	**0.898**	Daily	122.05 ± 12.98	**0.337**
4–6	121.62 ± 12.81		4–6	121.72 ± 13.59	
1–3	122.53 ± 13.43		1–3	123.34 ± 13.60	
No	123.96 ± 13.52		No	124.31 ± 13.24	

Different letters means that there is a statistically significant difference between the groups.

**Table 3 jpm-11-00873-t003:** Multivariate logistic regression between adolescent’s systolic hypertension and anthropometric measurements.

Multivariate Analysis						
Variables	Pre-SHTN ^‡^ OR (95% CI)	*p*-Value	Stage 1 SHTN ^‡^ OR (95% CI)	*p*-Value	Stage 2 SHTN ^‡^ OR (95% CI)	*p*-Value
**Body Mass Index ***						
Overweight	2.3 (1.4, 3.8)	**0.001**	1.8 (1.01, 3.1)	**0.047**	3.2 (2.1, 5.1)	**<0.001**
Obesity	2.7 (1.6, 4.5)	**<0.001**	2.7 (1.6, 4.6)	**<0.001**	7.1 (4.7, 10.9)	**<0.001**
**Waist circumferences**						
≥90th percentile	2.3 (1, 5.5)	**0.049**	5 (2.3, 10.6)	**<0.001**	8.6 (4.5, 16.4)	**<0.001**
**Total Fat †**						
Over fat	1.5 (0.8, 3.1)	**0.209**	1.1 (0.5, 2.4)	0.757	1.9 (1, 3.3)	**0.032**
Obese	2 (1.2, 3.4)	**0.009**	2.2 (1.3, 3.8)	**0.004**	3.5 (2.3, 5.5)	**<0.001**
**Trunk Fat**						
≥90th percentile	1.6 (0.8, 3.2)	0.195	4 (2.2, 7.1)	**<0.001**	5.5(3.4, 9.1)	**<0.001**
**Visceral Fat**						
≥90th percentile	1.8 (0.7, 4.6)	0.237	4.2 (1.9, 9.4)	**<0.001**	7.2 (3.7, 14.1)	**<0.001**

* BMI for age was obtained and categorized according WHO cutoff point underweight (for age BMI < 5% percentile), normal weight (5% percentile < BMI < 85% percentile), overweight (85% percentile < BMI < 95% percentile), obese (BMI ≥ 95% percentile) (Riley and Bluhm, 2012). ^†^ TOTAL FAT cut off point were determined. ^‡^ SHTN refer to systolic hypertension.

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
