# Peer review of "Identification of Adolescents with Adiposities and Elevated Blood Pressure and Implementation of Preventive Measures Warrants the Use of Multiple Clinical Assessment Tools"

_jpm, 2021, doi:10.3390/jpm11090873_

Round 1

Reviewer 1 Report

Thank you for allowing me to review your interesting manuscript. The topic is of great importance. The study was well executed and the science looks solid but the scope of the work needs to be better defined.

 I have several suggestions:

TITLE

The title now reads 'The association between abdominal adiposity and elevated blood pressure during adolescence using multiple assessment tools is confirmed'. I suggest this is replaced by (for example) 'Identification of adolescents with adipositas and elevated blood pressure and implementation of preventive measures warrants the use of multiple clinical assessment tools'.

I would suggest the paper also focusses on this aim on top of just reporting the data of the findings. 

 ABSTRACT

Please also see my note on the title.

The abstract should be structured (e.g. 'aims/introduction, methods and means, results and conclusion(s))

The abstract ends with results and not with the conclusions of the findings. It is suggested the concluding sentence identifies the need for use of multiple clinical assessment tools as well as the need for preventive measures in adolescents with overweight and elevated BP.

BODY OF MANUSCRIPT

Please also see my note on the title.

The authors state that there is a need for information on adipositas and (raised) BP in Jordan. However, their work is more extensive and is aimed at diagnoses and prevention to improve population health.

What the authors show is not just the need for more data, but also, that (little) extra effort on the workfloor by implementing a panel of assessment tools greatly optimizes identification of those at risk on population and individual level, as well as improves preventive measures on popultion and induvidual scales.   

here is a need for identification and specification of risk and precision of preventive treatment. The use of multiple tools must be emphasized in the concluding sentences of the paper and be synchronized with the text of the abstract.

This approach may merit the value of the paper as well as put the results into a practical scope; this may be part of the discussion section. Identical to the abstract, for the concluding sentences it may be sugggested these identifiy the need for use of multiple clinical asssessment tools as well as the need for preventive measures in adolescents with overweight and elevated BP. 

This information could be added in one or two short sentences amongst the already well presented text that emphasizes the clinical relevance of your data in the concluding sentences (page 10, line 249 - 253).

Otherwise, very interesting and relevant report.

Author Response

The responses are enclosed in the file attached. 

Reviewer 2 Report

The topic is actual because childhood obesity is one of the main public health issues. The manuscript is adequately addressed through literature evidence. This paper provides useful information’s that elevated BP was significantly associated with total and abdominal adiposity among adolescent males in Jordan. The discussion is well balanced and includes relevant literature data. However, there are some points which need to be addressed.

Major comment:

In Data collection authors said “All adolescents were asked to complete a self-administered questionnaire included data about socio-demographic status, dietary habits, and physical activity level “ Authors must present dietary data and physical activity levels and discuss according to adolescent obesity and BP. It is necessary to show the intake of food by food groups and make comments 

Minor comments

Line 44 Abstract … this study…  capital letter at the beginning of the sentence

Line 55 Key words   add males or boys

Line 115 insert space between TF% and were used

Line 159  instead found at Table 1 put presented in Table 1

Author Response

(The authors gave the same response as above.)

Round 2

Reviewer 2 Report

The authors responded to most of my comments.